Urate-lowering effects of polyphenolic compounds in animal models: systematic review and meta-analysis

Chen Jianhong
Zhang Boye
Cao Zhongzhi
Yang Li
Yuan Ye yuanye@cdutcm.edu.cn
Hospital of Chengdu University of Traditional Chinese Medicine , Chengdu , China
Menini Stefano
Electronic publication date: 2025 Aug 11
Publication date: 2025
Volume: 13
Electronic Location ID: e19731
Received 2025 Feb 27; Accepted 2025 Jun 19
Copyright: ©2025 Chen et al.
Copyright year: 2025
Copyright holder: Chen et al.
License: This is an open access article distributed under the terms of the Creative Commons Attribution License, which permits unrestricted use, distribution, reproduction and adaptation in any medium and for any purpose provided that it is properly attributed. For attribution, the original author(s), title, publication source (PeerJ) and either DOI or URL of the article must be cited.
License URL: https://creativecommons.org/licenses/by/4.0/

Keywords: Polyphenols, Uric acid, Animal model

Funding: Joint Innovation Fund of Health Commission of Chengdu Chengdu University of Traditional Chinese Medicine WXLH202403117 This Study was funded by Joint Innovation Fund of Health Commission of Chengdu and Chengdu University of Traditional Chinese Medicine, WXLH202403117. The funders had no role in study design, data collection and analysis, decision to publish, or preparation of the manuscript.

==============================
Background

Recent research underscores the critical role of uric acid (UA) in the pathogenesis and progression of various diseases. However, the effects of polyphenolic compounds on uric acid levels remain poorly defined.

Objective

This review aims to assess the impact of five specific polyphenolic compounds on uric acid levels in animal models.

Methodology

We performed an exhaustive literature search through October 30, 2024, utilizing databases including Wanfang, VIP, Cochrane Library, CNKI, Embase, and PubMed. The methodological quality of the included animal studies was evaluated using the SYRCLE (Systematic Review Centre for Laboratory animal Experimentation) risk of bias tool. Data analysis was conducted using R software, with meta-analyses performed via RevMan 5.3, adhering to the Cochrane Handbook for Systematic Reviews of Interventions.

Results

Our analysis integrated data from 49 studies, revealing that the selected polyphenolic compounds significantly lowered serum uric acid (SUA) levels across various animal models (standardized mean difference (SMD) = −2.33, 95% CI [−2.73, −1.93]) and increased urinary uric acid (UUA) levels (SMD = 2.53, 95% CI [1.38, 3.69]). Subgroup analyses demonstrated consistent SUA reduction across different disease models. Detailed meta-analyses for each polyphenol disclosed distinct contributions to SUA reduction: resveratrol (RES) (SMD = −1.86, 95% confidence interval (CI) [−2.28, −1.45]), chlorogenic acid (CGA) (SMD = −2.31, 95% CI [−2.89, −1.73]), ferulic acid (FA) (SMD = −2.82, 95% CI [−4.46, −1.19]), punicalagin (PU) (SMD = −3.87, 95% CI [−5.99, −1.75]), and bergenin (BER) (SMD = −8.51, 95% CI [−10.30, −6.73]).

Conclusion

This meta-analysis supports the proposition that polyphenols such as RES, CGA, FA, PU, and BER effectively reduce serum uric acid in animal models. Notably, RES exhibited an inverted U-shaped nonlinear trend. However, the high heterogeneity and methodological constraints, including small sample sizes, ambiguous randomization practices, and potential publication bias, necessitate cautious interpretation. Further high-quality research is essential to substantiate these findings and facilitate their translation into clinical practice.

Introduction

Uric acid (UA) is a sparingly water-soluble weak organic acid capable of undergoing acid–base neutralization reactions with bases, and exhibits a certain degree of reductivity. This property is fundamental to its role in inducing various physiological changes. Elevated serum uric acid (SUA) levels lead to the precipitation of UA, which is minimally soluble, resulting in the formation of urate crystals. These crystals can cause physical damage to vascular endothelium. Additionally, phagocytic cells within the bloodstream may ingest these crystals, recognizing them as foreign bodies, thereby releasing inflammatory mediators and triggering inflammatory responses (Choi et al., 2014; Goldberg et al., 2021; Zu et al., 2022). Research has demonstrated that high concentrations of UA increase oxidative stress within the body, leading to the production of substantial amounts of reactive oxygen species (ROS) that cause extensive cellular damage (Tian et al., 2023). In humans, UA is primarily a byproduct of purine metabolism, with the majority being excreted through the kidneys and a smaller portion through the gastrointestinal tract (Peters & Woodward, 2019). Under normal circumstances, a balance between production and excretion maintains SUA levels within a stable range. However, certain pathologies can disrupt this balance by either increasing UA production or decreasing its excretion, thus elevating SUA levels beyond the normal range. For instance, glomerulonephritis can impede UA excretion, while acute lymphocytic leukemia can lead to excessive UA production (Lorie Iu & Poddubnaia, 1970). Moreover, conditions such as diabetes and hyperlipidemia may elevate UA levels, which may precipitate further diseases. Specifically, the deposition of urate crystals can initiate gout, and UA accumulation in the kidneys can lead to uric acid nephropathy, ultimately resulting in renal failure (Huang et al., 2020). Excessively high SUA levels may also damage vascular endothelium, induce thrombosis, and lead to cardiovascular diseases (Saito et al., 2021). Thus, primary hyperuricemia can cause severe diseases, and secondary hyperuricemia can lead to a cascade of serious complications. Therefore, controlling elevated UA levels is critical in managing the diseases associated with it.

Current clinical therapies frequently utilize urate-lowering drugs. Allopurinol, which inhibits UA production, may cause severe allergic reactions; febuxostat is linked to liver damage and associated with cardiovascular risks. Probenecid, which reduces UA reabsorption and increases UA excretion, may lead to excessively high urinary uric acid levels, resulting in crystal formation within the urinary system and thereby causing damage, as well as potential gastrointestinal reactions (Cai, 2010; Dalbeth & Stamp, 2015; Juan et al., 2018; Yu Guoming, 2019). As a result, the limitations of current urate-lowering medications may prevent some patients from using them, leading to complications associated with elevated UA levels. Hence, the development of new and effective urate-lowering drugs is imperative.

Recent studies have increasingly recognized the wide array of health benefits attributed to polyphenols, particularly in mitigating cardiovascular diseases, neurodegenerative disorders, obesity, and diabetes (Cory et al., 2018; Dudkowiak, Gryglas & Poniewierka, 2016; Fraga et al., 2019; Luca et al., 2020; Pelczyńska et al., 2022; Rana et al., 2022; Rasouli, Farzaei & Khodarahmi, 2017; Russo et al., 2020). Scholarly discourse has previously identified a correlation between uric acid (UA) levels and these health conditions, proposing that the advantageous impact of polyphenols may partly stem from their capacity to reduce UA levels. Distinct from common polyphenols such as quercetin and catechins, which solely inhibit xanthine oxidase (XO), resveratrol (RES) is unique in exhibiting dual inhibitory effects. It not only hampers XO activity but also suppresses the expression of the uric acid transporter URAT1 in renal tubules, thereby concurrently decreasing uric acid synthesis and facilitating its excretion (Lee et al., 2017; Zhou et al., 2023). Chlorogenic acid (CGA) is distinguished by its gut-kidney synergistic regulation and robust intestinal stability, which contribute to its uric acid-lowering effect. By modulating the metabolic pathways of intestinal flora and inhibiting hepatic XO activity, along with its quinic acid component binding specifically to the renal organic anion transporter OAT1, CGA promotes uric acid excretion (Zhou et al., 2021; Zhou et al., 2022). Ferulic acid (FA) has been demonstrated to inhibit xanthine oxidase activity in vitro and in vivo, exhibiting superior efficacy to coumaric acid and chlorogenic acid with reduced hepatorenal toxicity. Furthermore, it activates the Nrf2/ARE pathway to upregulate synthesis of glutathione reductase and quinone oxidoreductase (NQO1), thereby enhancing cellular antioxidant defenses. Given the duality of uric acid which functions as an antioxidant at physiological concentrations but promotes oxidative stress in hyperuricemic states, the potential direct modulation of uric acid homeostasis by this pathway remains unresolved (Sun et al., 2025; Xu, Zhao & Xiao, 2020). Punicalagin (PU), a prominent ellagitannin found in pomegranate, is noted for its regulation of renal glucose metabolism to alleviate renal burden and for remodeling gut microbiota to promote uric acid catabolism (Han et al., 2025). Moreover, recent studies have demonstrated that Bergenin (BER) exhibits a significant dual (renal and intestinal) uric acid excretion effect and possesses anti-inflammatory properties, including the promotion of macrophage polarization from the M1 to the M2 phenotype (Chen et al., 2020). Furthermore, meta-analyses provide additional insights. One analysis indicated that quercetin effectively reduced uric acid by modulating various signaling pathways, including oxidative stress and lipid metabolism in animal models of hyperuricemia (Bian et al., 2025). However, another analysis suggested that while quercetin significantly alleviated renal inflammation in an animal model of diabetic nephropathy, it did not markedly affect uric acid levels, possibly due to variations in the types of nephropathy present in the models (Hu et al., 2022). In light of these findings, this study focuses on five polyphenols—RES, CGA, FA, PU, and BER—each of which has been extensively represented in contemporary research, to further explore their effects on UA management.

However, comprehensive studies on the isolated effects of these five polyphenols on UA reduction and their mechanisms are still lacking. Given the limited number and size of clinical studies on the effects of these polyphenols on UA, most evidence currently relies on animal experiments, which show varying results. There is a lack of systematic analysis of the mechanisms behind their urate-lowering effects, and direct evidence of their efficacy and mechanisms in humans remains scarce. We plan to carry out a comprehensive review and meta-analysis of relevant studies involving animals for exploring whether polyphenols generally possess UA-lowering properties and for investigating the effects and potential mechanisms of these purified compounds on UA, thereby providing direction and evidence-driven support in developing new medications to reduce urate.

Materials and Methods

Protocol

This systematic review and meta-analysis will be conducted in strict accordance with the protocol registered with PROSPERO (CRD42024617024) and will adhere to the PRISMA guidelines (Moher et al., 2009).

Literature search strategy

The search covered databases from their inception until October 30, 2024. Databases searched include Embase, Wanfang database of Chinese academic institutions, PubMed, VIP database of Chinese technical journals, China National Knowledge Infrastructure (CNKI), and the Cochrane Library. The basic search framework involved terms for (“resveratrol”), (“chlorogenic acid”), (“ferulic acid”), (“punicalagin”), (“bergenin”), along with “ (serum uric acid)” and “(urinary uric acid)”. Details of the search strategy are provided in Supplementary Material 1.

Inclusion criteria

1. Study Type: in vivo animal experiments.

2. Subjects: animal models without restrictions on type, breed, gender, weight, or age.

3. Intervention: the experimental group received one of the polyphenols (RES, CGA, FA, PU, BER), while the control group received either a vehicle or a disease-inducing agent.

4. Outcomes: the primary outcome is serum uric acid, and the secondary outcome is urinary uric acid.

Exclusion criteria

1. Studies lacking customs-related data.

2. Studies with duplicate publications (retaining the study with the more comprehensive data set).

3. Research employing plant extracts containing the aforementioned five types of polyphenolic compounds or used in conjunction with other formulations.

4. Review articles.

5. In vitro experiments.

6. Conference abstracts.

7. Clinical studies.

8. Case reports.

Data extraction and management

Literature screening was independently conducted by two researchers, who systematically reviewed titles, abstracts, and full texts to identify studies meeting the predefined inclusion criteria. Discrepancies were resolved by consulting a third researcher. After preliminary screening, detailed examination of the full texts was undertaken based on the inclusion and exclusion criteria for further evaluation and data extraction. Data were independently extracted into a structured Excel database, capturing variables including author(s), year of publication, country of study, subject gender, species, age, body weight, dosage administered, model type, model induction pathway, details of experimental and control groups, intervention strategies, administration route, duration of intervention, and outcome measures. For all outcomes, data were extracted including sample sizes (N), means, standard deviations (SD), or standard errors of the mean (SEM). To standardize variability measures, the formula SD = SEM ×N was employed to convert SEM values into SD. Outcome data were consistently recorded as means with their respective standard deviations. In instances where studies featured multiple experimental groups and a single control group, data from the control group were appropriately allocated across different comparisons. Each paired comparison was then included in the meta-analysis (Higgins et al., 2024). When study results were presented solely in graphical format, data extraction was performed using WebPlotDigitizer, and this extraction process was replicated three times to ensure precision. Efforts were made to contact authors for missing essential data; studies were excluded if authors did not respond, resulting in incomplete critical data.

Quality assessment of literature

The risk of bias within the included animal studies was evaluated by two researchers using the SYRCLE tool (Zeng et al., 2015). This assessment covered ten critical domains:

1. Sequence generation.

2. Baseline characteristics.

3. Allocation concealment.

4. Random housing.

5. Blinding of caregivers and investigators.

6. Random outcome assessment.

7. Blinding of outcome assessors.

8. Reporting of incomplete data.

9. Selective outcome reporting.

10. Other sources of bias.

Every study included was evaluated on an individual basis using these standards. A study was deemed “low risk” if it met all the criteria completely, reflecting high quality and a low probability of bias. Studies that met the criteria only partially were labeled as “unclear risk”, implying a moderate chance of bias. Those that did not meet the criteria at all were rated as “high risk”, signaling greater bias and reduced quality. Any conflicts were settled through consultation with a third researcher.

Statistical analysis

Statistical and meta-analytic computations were performed using R software version 4.4.0 and Review Manager (RevMan) 5.3. Outcomes were treated as continuous variables. When the studies shared uniform units of effect size and methods of measurement, the effect sizes were expressed using the 95% confidence intervals (CI) and weighted mean difference (WMD). In cases of non-uniformity, the standardized mean difference (SMD) and 95% CI were utilized instead. Heterogeneity among included studies was assessed using the I2 statistic or the Q test. An I2 value of 0%, 25%, 50%, and 75% indicated no, low, moderate, and high heterogeneity, respectively. For I2 values ≥50%, sensitivity analyses were conducted by sequentially removing individual studies to observe changes in the I2 value, exploring potential sources of heterogeneity, and assessing the stability of the results. If heterogeneity could not be addressed, a random effects model was employed along with subgroup analysis. Conversely, if the I2 value was <50%, a fixed effects model was applied. For meta-analyses comprising ten or more studies, funnel plots and Egger’s test were conducted to assess publication bias, with a significance threshold set at p < 0.05 (Egger et al., 1997).

Results

Included studies

Following the specified search strategy, a total of 1,060 articles were retrieved from six databases: PubMed (145 articles), Cochrane Library (11 articles), Embase (354 articles), CNKI (462 articles), VIP (20 articles), and Wanfang (68 articles). After the removal of duplicates, 266 articles were excluded. Screening of titles and abstracts led to the exclusion of 687 irrelevant publications, including reviews, reports, and clinical studies. Upon full-text review, 58 additional articles were excluded, leaving 49 articles for inclusion (Adikwu, Biradee & Ogungbaike, 2019; Alam, Sernia & Brown, 2013; Alhusaini et al., 2024; Anila et al., 2022; Bagul et al., 2018; Bagul et al., 2012; Bai et al., 2021; Che, 2024; Chen, 2013; Chowdhury et al., 2022; Da Silva et al., 2022; Erseçkin et al., 2022; Fan et al., 2009; Fang, 2015; Ferraz-Filha et al., 2017; Han et al., 2025; Hsu et al., 2020; Kaur et al., 2016; Lee et al., 2017; Li et al., 2021; Xu et al., 2019; Martins de Sá Müller et al., 2019; Meng et al., 2014; Palsamy & Subramanian, 2008; Peng et al., 2012; Qinglian, 2023; Rai, Bagul & Banerjee, 2020; Ren, 2015; Salau et al., 2023; Sammeturi et al., 2019; Sanjeev et al., 2019; Sarkar et al., 2019; Shi et al., 2012; Singh et al., 2021; Song, 2019; Sun Yadi et al., 2023; Wang, 2023; Xu et al., 2024; Xue et al., 2017; Yuan, 2017; Zhang et al., 2019; Zhang et al., 2022; Zhang et al., 2021; Zhang et al., 2017; Zhao et al., 2019; Zhao et al., 2016; Zhou & Chen, 2014; Zhou et al., 2024; Zhou et al., 2021). Two studies by Fan et al. (2009) and Alam, Sernia & Brown (2013) were further divided into four separate groups due to differences in modeling methodologies, animal species, and intervention durations, resulting in Fan(1)2009, Fan(2)2009, Alam(1)2013, and Alam(2)2013. A detailed flowchart of the literature screening process is depicted in Fig. 1.

Figure 1 PRISMA 2020 flow diagram.

Characteristics of included studies

The meta-analysis incorporated 49 studies conducted between 2008 and 2024, involving 81 independent control groups and 1,252 experimental animals. This cohort included 742 rats (517 Sprague-Dawley rats, 191 Wistar rats, 20 Albino rats, and 14 spontaneously hypertensive rats (SHR)) and 510 mice (318 Km mice, 69 Swiss mice, 107 C57BL/6J mice, and 16 ICR mice). Sprague-Dawley rats were utilized in 20 studies, Wistar rats in 10 studies, Albino rats in one study, and SHRs in one study. Km mice were used in nine studies, Swiss mice in five studies, C57BL/6J mice in four studies, and ICR mice in one study. The dosage range for polyphenolic compounds spanned from 5 to 1,000 mg/kg. Ten studies investigated three different dosages, another ten studies two dosages, and the remaining 29 studies focused on a single dosage. RES was used in 29 studies, FA in nine, CGA in eight, PU in two, and BER in one. Administration routes included oral gavage in 26 studies, oral administration in 18 studies, and intraperitoneal injection in five studies. The duration of intervention with polyphenolic compounds varied from 6 h to 20 weeks. Detailed characteristics of the included studies are presented in Table 1.

Table 1 Basic characteristics of the included studies.

Author	Study year	Animal type	Country	Types of intervention	Age (week)	Sex	Weight (g)	Inducer	Drug route	Intervention duration (d)	Dose	Sample size (EG/CG)	Outcomes	Disease models	
Palsamy & Subramanian	2008	Wistar rat	India	RES	–	Male	160–180	Diabetes was induced by a single intraperitoneal injection of streptozotocin (50 mg/kg) dissolved in 0.1 M cold citrate buffer (pH 4.5). Rats were intraperitoneally injected with nicotinamide (110 mg/kg) 12 h after injection.	p.o	30	5 mg/kg	6/6	SUA	Diabete	
Fan et al. (1)	2009	Swiss mice	China	RES	–	Male	25–35	CCl4 (1.0 ml) mixed with olive oil (4 ml) p.o.	p.o	1	30 mg/kg	6/6	SUA	Acute liver injury	
Fan et al. (2)	2009	Swiss mice	China	RES	–	Male	25–35	CCl4 (1.0 ml) mixed with olive oil (4 ml) s.c.	p.o	1	30 mg/kg	6/6	SUA	Acute liver injury	
Shi et al.	2012	Km mice	China	RES	–	Male	18–22	Mice were given 15 mL/kg oxonic acid (250 mg/kg) or water (solvent) by intragastric administration at 8:00 AM once daily for 7 days	p.o	7	20–40 mg/kg	8/8/8	SUA–UUA	HUA	
Peng et al.	2012	SD rat	China	FA	4	Male	155(150–164)	DR (Pfizer, Milano, Italia) 8.5 mg/kg was injected subcutaneously	p.o	84	70 mg/kg	6/6	SUA	DR-CKD	
Bagul et al.	2012	SD rat	India	RES	–	Male	180–200	The 65% fructose diet was maintained for 56 days	p.o	56	10 mg/kg	8/8	SUA	Diabete	
Chen	2013	Km mice	China	RES	6	Male	18–22	25% yeast extract diet plus potassium oxonate i.p. 250 mg/kg	i.g.	14	20 mg/kg	10/10	SUA	HUA	
Alam, Sernia & Brown (1)	2013	Wistar rat	Australia	FA	8–9	Male	325–327	L-NAME (50 mg/kg/day) was dissolved in water and fed for 8 weeks	p.o	28	50 mg/kg	8/8	SUA	Hypertensive	
Alam, Sernia & Brown (2)	2013	9SHR	Australia	FA	10	Male	407–411	spontaneously hypertensive rats	p.o	84	50 mg/kg	7/7	SUA	Hypertensive	
Zhou & Chen	2014	Km mice	China	BER	–	Male	18–22	Potassium oxonate solution i.g. 250 mg/kg	i.g.	7	20–40–60 mg/kg	10/10/10/10	SUA–UUA	HUA	
Meng et al.	2014	Km mice	China	CGA	–	Male	18–22	Intravenous PO (250 mg/kg, 20 mL/kg) was given at 8 am daily for 7 consecutive days	p.o	7	50–100–200 mg/kg	12/12/12/12	SUA–UUA	HUA	
Ren	2015	SD rat	China	RES	–	Male	220–240	Potassium oxonate i.g. 700 mg/kg	i.g.	56	20–40 mg/kg	10/10/10	SUA–UUA	HUA	
Fang	2015	SD rat	China	RES	–	Male	140–160	High yeast high fat diet feeding + oxonic acid milk suspension i.g. 250 mg/kg	i.g.	84	100 mg/(kg⋅d)	12/12	SUA	HUA and NAFLD	
Zhao et al.	2016	SD rat	China	RES	–	Male	140–160	High yeast and high fat diet feeding + potassium oxonate milk suspension Sc 100 mg/(kg⋅d)	i.g.	84	100 mg/(kg⋅d)	12/12	SUA	HUA and NAFLD	
Kaur et al.	2016	SD rat	India	RES	–	Male	200–250	Single i.p. injection of streptozotocin (50 mg/kg)	i.g.	28	25 mg/kg	8/8	SUA	Diabete	
Zhang et al.	2017	SD rat	China	RES	8–12	Female	200–240	STZ i.p. 35 mg⋅kg	i.g.	14	60–120–240 mg/kg	16/16/16/16	SUA	GDM	
Yuan	2017	SD rat	China	RES	7	Male	180–220	The rat model of renal I/R injury was established by stripping and clamping the bilateral renal pedicle vessels for 45 min and then releasing the clips	i.p.	14	10–20–40 mg/(kg⋅d)	20/20/20/20	SUA	RIRI	
Zhang et al.	2017	SD rat	China	RES	–	Male	200	i.g. adenine (200 mg/kg) for 1 month	i.g.	60	10–15–20 mg/kg	5/5/5/5	SUA	CKD	
Lee et al.	2017	SD rat	China	RES	–	Male	160–180	200 mg/kg potassium oxybate and 50 mg/kg UA mixed i.g.	i.p.	21	10 mg/kg	8/8	SUA	HUA	
Ferraz-Filha et al.	2017	Swiss mice	Brazil	CGA	–	Male	25–30	i.p. oxonate potassium 250 mg/kg on days 1 and 3	i.g.	3	10–15 mg/kg	6/6/6	SUA	HUA	
Bagul et al.	2018	SD rat	India	RES	8–10	Male	200–220	STZ 50 mg/kg, pH 4.5, prepared in a single i.p. citrate buffer	p.o	56	25 mg/kg	12/12	SUA	Diabete	
Zhang et al.	2019	Km mice	China	CGA	–	Male	20–25	Hypoxanthine i.p. 1000 mg/kg	i.g.	7	5 mg/kg	10/10	SUA	HUA	
Xu et al.	2019	C57BL/6J mice	China	RES	6	Male	18–20	The high-fat diet was fed for 126 days	i.g.	126	5 mg/(kg⋅d)	6/6	SUA	HL	
Song	2019	SD rat	China	RES	–	Male	140–160	High yeast high fat diet was fed for 84 days + oxonic acid milk suspension 250 mg/kg	i.g.	84	100 mg/kg.d	12/12	SUA	HUA and NAFLD	
Zhao et al.	2019	SD rat	China	FA	–	Male	180–220	Renal calculi were induced by adding ethylene glycol (0.75%v/v) to drinking water	p.o	28	40–80 mg/kg	6/6/6	SUA–UUA	Renal calculus	
Sarkar et al.	2019	Wistar rat	India	RES	–	Male	150–180	55 mg/kg streptozotocin i.p. (dissolved in 0.1 mg cold citrate buffer, pH 4.5)	p.o	30	40 mg/kg	5/5	SUA	Diabete	
Sanjeev et al.	2019	Wistar rat	India	FA	12	Male	180–190	10 mg/kg cadmium chloride, dissolved in 0.9% saline, was injected subcutaneously for 15 and 30 days	p.o	15–30	50 mg/kg	10/10	SUA	Hepatic and renal damage	
Sammeturi et al.	2019	Wistar rat	India	RES	–	Male	100–120	ISO (isoproterenol) [50 mg/kg bw]: ISO was administered subcutaneously on days 29 and 30 with an interval of 24 h	p.o	30	50 mg/kg	6/6	SUA	Myocardial damage.	
Adikwu, Biradee & Ogungbaike	2019	Albino rats	Nigeria	RES	–	Male	200–250	5-FU i.p. 20 mg/kg, once a day for 5 days	i.p.	5	10–20–40 mg/kg	5/5/5/5	SUA	5-fluorouracil- induced nephrotoxicity	
Martins de Sá Müller et al.	2019	Swiss mice	Brazil	CGA	–	Male	25–30	i.p. oxonate potassium 250 mg/kg on days 1 and 3	i.g.	3	15 mg/kg	6/6	SUA	HUA	
Hsu et al.	2020	ICR mice	China	RES	48	Male	–	Mass-drop injury	i.g.	7	25 mg/kg	8/8	SUA	MDI	
Rai, Bagul & Banerjee	2020	SD rat	India	RES	–	Male	180–200	65% fructose diet for 140 days	p.o	140	10 mg/kg	8/8	SUA	Diabete	
Bai et al.	2021	SD rat	China	RES	6	Male	180–220	750 mg/kg potassium oxonate i.g.	i.g.	35	20 mg/kg	8/8	SUA	HUA	
Zhou et al.	2021	Km mice	China	CGA	–	Male	18–22	HX i.g. 300 mg/kg+PO i.p.300 mg/kg	p.o	19	30–60 mg/kg	10/10/10	SUA	HUA	
Zhang et al.	2021	C57BL/6J mice	China	RES	6	Male	21–23	High-fat diet was fed for 42 days	i.g.	42	100 mg/kg	10/10	SUA	HFD- induced insulin resistance (IR)	
Li et al.	2021	Km mice	China	RES	–	Male	18–22	D-gal solution i.p (100 mg kg−1 d−1)	i.g.	28	10–20–40 mg/kg	8/8/8/8	SUA–UUA	Subacute aging	
Singh et al.	2021	Wistar rat	India	CGA	–	Either sex	150–200	STZ i.p.(50 mg/kg)	p.o	28	100–150 mg/kg	5/5/5	SUA	T2D	
Zhang et al.	2022	SD rat	China	FA	8	Male	–	The rats were fed with high-fat, high-fructose (HFFD) diet for 140 days	p.o	140	34.5–69 mg/kg	10/10/10	SUA	MetS	
Da Silva et al.	2022	Swiss mice	Brazil	CGA	6-8	Either sex	25–35	i.p. BLV 50µg/mouse	i.p.	0.25	10–20 mg/kg	5/5/5	SUA	BLV Injure	
Erseçkin et al.	2022	Wistar rat	Turkey	FA	7–8	Female	250–300	i.p. Gentamicin80 mg/kg/d	i.p.	8	50 mg/kg	8/8	SUA	Gentamicin- induced nephrotoxicity	
Chowdhury et al.	2022	Wistar rat	Bangladesh	RES	10–12	Male	185–200	They were fed with a high-fat diet for 56 days	i.g.	56	100 mg/kg	7/7	SUA	HF diet-fed	
Anila et al.	2022	Wistar rat	India	RES	–	Either sex	200–250	They were fed a high-fructose diet for 35 days	p.o	35	20 mg/kg	30/30	SUA	Fructose induced MS	
Sun Yadi et al.	2023	Km mice	China	FA	9	Male	18–22	2 mg/kg cadmium chloride solution i.g.	i.g.	14	75–150–300 mg/kg	16/16/16/16	SUA	AKI (cadmium- induced)	
Wang	2023	C57BL/6J mice	China	RES	7	Male	20.5–21.5	Potassium oxonate (PO) (250 mg/kg), adenine (100 mg/kg), yeast (1 g/kg)	i.g.	21	100 mg/kg	15/15	SUA	HUA	
Hua	2023	C57BL/6J mice	China	PU	8	Male	22–23	D12492 was fed with a high-fat diet for 56 days	i.g.	56	50–100 mg/kg	15/15/15	SUA	DKD	
Salau et al.	2023	SD rat	South Africa	FA	–	Male	224 ± 13.62	i.p.40mg/kg of streptozotocin (STZ)	i.g.	35	150–300 mg/kg	5/5/5	SUA	T2D	
Xu et al.	2024	SD rat	China	RES	–	Male	180–220	Gouty nephropathy was induced by potassium oxonate combined with sodium urate	i.g.	7	250–500–1000 mg/kg	10/10/10/10	SUA	GRI	
Che	2024	SD rat	China	FA	6	Male	200	Oxonate potassium salt solution i.g. 100 mg/kg	i.g.	56	125 mg/kg	8/8	SUA	HUA	
Zhou et al.	2024	SD rat	China	CGA	6	Male	180–220	nephrectomy+potassium oxonate i.p.	i.g.	7	300 mg/kg	8/8	SUA	CKD	
Alhusaini et al.	2024	Wistar rat	Saudi Arabia	RES	–	Male	150–180	Rats received ISO at a dose of 50 mg/kg (sc) twice weekly for two weeks	p.o	14	20 mg/kg	8/8	SUA	ISO-induced kidney injury	
Han et al.	2025	Km mice	China	PU	–	Male	18–22	PO i.g. 300 mg/kg/d+AD i.g. 200 mg/kg/d	i.g.	14	100–200–300 mg/kg	10/10/10/10	SUA	HUA	

Quality assessment of studies

All studies were assessed as having unclear risk regarding baseline characteristics, allocation concealment, blinding of animal caregivers and researchers, and random outcome assessment. Only two studies provided detailed descriptions of random sequence generation, and 36 mentioned randomization in animal housing. One study noted that outcome assessors were blinded. Six studies reported incomplete data, as outcomes were not fully reported according to the experimental design. The risk of selective reporting and other potential biases was deemed low. Summaries and graphical representations of bias risks are shown in Figs. 2 and 3.

Figure 2 Risk of bias plot.

Figure 3 Summary of risk of bias.

Effects of polyphenolic compounds on SUA

Impact of five polyphenolic compounds on SUA

A comprehensive review of fifty-one experiments, derived from forty-nine publications and involving 1,252 experimental animals, explored the effects of five distinct polyphenolic compounds on SUA levels in various animal models. The synthesized data indicated that these compounds generally lead to a reduction in SUA levels, albeit with significant variability among the results (SMD = −2.33, 95% CI [−2.73, −1.93], p < 0.0001, I2 = 87%) (Fig. 4). A random effects model was utilized to manage the variability, and a subsequent sensitivity analysis corroborated the robustness of these findings (Fig. S1).

Figure 4 Forest plot of SUA after five polyphenol interventions.

Notes: Palsamy & Subramanian, 2008; Fan et al., 2009; Shi et al., 2012; Peng et al., 2012; Bagul et al., 2012; Chen, 2013; Alam, Sernia & Brown, 2013; Zhou & Chen, 2014; Meng et al., 2014; Ren, 2015; Fang, 2015; Zhao et al., 2016; Kaur et al., 2016; Zhang et al., 2017; Yuan, 2017; Lee et al., 2017; Ferraz-Filha et al., 2017; Bagul et al., 2018; Zhang et al., 2019; Xu et al., 2019; Song, 2019; Zhao et al., 2019; Sarkar et al., 2019; Sanjeev et al., 2019; Sammeturi et al., 2019; Adikwu, Biradee & Ogungbaike, 2019; Martins de Sá Müller et al., 2019; Hsu et al., 2020; Rai, Bagul & Banerjee, 2020; Bai et al., 2021; Zhou et al., 2021; Zhang et al., 2021; Zhang et al., 2017; Li et al., 2021; Singh et al., 2021; Zhang et al., 2022; Da Silva et al., 2022; Erseçkin et al., 2022; Chowdhury et al., 2022; Anila et al., 2022; Sun Yadi et al., 2023; Wang, 2023; Hua, 2023; Salau et al., 2023; Xu et al., 2024; Che, 2024; Zhou et al., 2024; Alhusaini et al., 2024; Han et al., 2025; Xue et al., 2017.

Further, subgroup analyses, stratified by animal species and specific disease models, demonstrated that the impact of polyphenols was markedly greater in Km mice (SMD = −5.56, p < 0.0001) and in models of liver disease (SMD = −5.56, p < 0.0001). This suggests a particularly strong efficacy of polyphenols in reducing SUA under these conditions. These subgroup findings also contributed to clarifying the sources of the noted heterogeneity (Table S1).

Impact of RES on SUA

An evaluation of thirty experiments, encompassing 732 experimental animals across twenty-nine publications, assessed the effects of RES on SUA levels in animal models. The pooled data from these studies reveal that RES significantly lowers SUA levels, although considerable heterogeneity was observed (SMD = −1.86, 95% CI [−2.28, −1.45], p < 0.0001, I2 = 82%) (Fig. 5). A random effects model was again applied, and sensitivity analysis validated the consistency of these results (Fig. S2).

Figure 5 Forest plot of SUA after RES interventions.

Notes: Palsamy & Subramanian, 2008; Fan et al., 2009; Shi et al., 2012; Bagul et al., 2012; Chen, 2013; Ren, 2015; Fang, 2015; Zhao et al., 2016; Kaur et al., 2016; Zhang et al., 2017; Yuan, 2017; Zhang et al., 2017; Lee et al., 2017; Bagul et al., 2018; Xu et al., 2019; Song, 2019; Sammeturi et al., 2019; Adikwu, Biradee & Ogungbaike, 2019; Hsu et al., 2020; Rai, Bagul & Banerjee, 2020; Bai et al., 2021; Zhang et al., 2021; Li et al., 2021; Chowdhury et al., 2022; Anila et al., 2022; Wang, 2023; Xu et al., 2024; Alhusaini et al., 2024.

Detailed subgroup analysis highlighted that significant reductions in SUA levels were most prominent when the RES dosage ranged from 26 mg/kg to 50 mg/kg (SMD = −2.72, p < 0.0001), the treatment duration spanned from four to eight weeks (SMD = −2.69, p = 0.0002), the administration route was oral (SMD = −2.38, p = 0.0003), and when Swiss mice were used as the model (SMD = −3.29, p < 0.0001). Interestingly, the urate-lowering effect of RES displayed an inverted U-shaped nonlinear trend, with efficacy peaking within the 26–50 mg/kg dosage range and diminishing at higher doses. These observations not only underscore the potential of RES to optimize urate-lowering under specific conditions, but also illuminate the underlying factors contributing to the observed heterogeneity (Table S1).

Impact of CGA on SUA

Eight studies involving 174 experimental animals reported on the effects of CGA on SUA levels in animal models. Results indicate that CGA significantly reduces SUA levels, albeit with substantial heterogeneity (SMD = −2.31, 95% CI [−2.89, −1.73], p < 0.0001, I2 = 57%) (Fig. 6). A random effects model was used, and sensitivity analysis confirmed the robustness of the results (Fig. S3).

Figure 6 Forest plot of SUA after CGA interventions.

Notes: Meng et al., 2014; Ferraz-Filha et al., 2017; Zhang et al., 2019; Martins de Sá Müller et al., 2019; Zhou et al., 2021; Singh et al., 2021; Da Silva et al., 2022; Zhou et al., 2024.

Subgroup analysis showed significant reductions in SUA levels when CGA dosage was between 101 mg/kg and 300 mg/kg (SMD = −3.81, p < 0.0001), the duration was 2 to 4 weeks (SMD = −2.86, p < 0.0001), and in models using SD rats (SMD = −4.38). These subgroup analyses help to explain some of the observed heterogeneity (Table S1).

Impact of FA on SUA

Eleven experiments from nine publications involving 221 experimental animals reported on the impact of FA on SUA levels. The included studies showed that FA significantly lowers SUA levels, but with high heterogeneity (SMD = −2.82, 95% CI [−4.46, −1.19], p = 0.0007, I2 = 93%) (Fig. 7). A random effects model was applied, and sensitivity analysis showed relatively stable results, although the exclusion of Sun2023 led to a notable change in the overall effect size, still within the original confidence interval (Fig. S4).

Figure 7 Forest plot of SUA after FA interventions.

Notes: Peng et al., 2012; Alam, Sernia & Brown, 2013; Zhao et al., 2019; Sanjeev et al., 2019; Zhang et al., 2022; Erseçkin et al., 2022; Sun Yadi et al., 2023; Salau et al., 2023; Che, 2024.

Subgroup analysis indicated significant reductions in SUA levels when FA dosage ranged from 101 mg/kg to 300 mg/kg (SMD = −9.97, p = 0.0001), the duration was 0 to 2 weeks (SMD = −18.93, p = 0.03), administered via oral gavage (SMD = −12.47, p < 0.0001), and in models using Km mice (SMD = −25.23, p < 0.0001). These subgroup analyses elucidate some sources of the observed heterogeneity (Table S1).

Impact of PU on SUA

Two studies involving 85 experimental animals reported the effects of PU on SUA levels in animal models. The pooled results demonstrate a significant reduction in SUA levels, albeit with high heterogeneity (SMD = −3.87, 95% CI [−5.99, −1.75], p = 0.0004, I2 = 92%) (Fig. 8). A random effects model was employed, and sensitivity analysis confirmed the robustness of these results (Fig. S5).

Figure 8 Forest plot of SUA after PU interventions.

Notes: Hua, 2023; Han et al., 2025.

Subgroup analyses indicated significant reductions in SUA levels with PU dosages between 101 mg/kg and 300 mg/kg (SMD = −7.93, p < 0.0001), durations of 0 to 2 weeks (SMD = −6.49, p < 0.0001), and in models using Km mice (SMD = −6.49, p < 0.0001). These subgroup analyses help to explain some of the observed heterogeneity (Table S1).

Impact of BER on SUA

A single study involving 40 experimental animals provided a descriptive analysis of the effects of BER on SUA levels in animal models. Compared to the control group, BER significantly reduced SUA levels (SMD = −8.51, 95% CI [−10.30, −6.73], p < 0.0001, I2 = 0%) (Fig. 9).

Figure 9 Forest plot of SUA after BER interventions.

Note: Zhou & Chen, 2014.

The effect of polyphenolic compounds on UUA

Six studies involving 192 experimental animals reported the effects of polyphenolic compounds on UUA levels in animal models. The results indicate that these compounds significantly increase UUA levels, exhibiting high heterogeneity (SMD = 2.53, 95% CI [1.38–3.69], p < 0.01, I2 = 91%) (Fig. 10). A random effects model was applied, and sensitivity analysis confirmed the robustness of the findings (Fig. S6).

Figure 10 Forest plot of UUA after five polyphenol interventions.

Notes: Shi et al., 2012; Zhou & Chen, 2014; Meng et al., 2014; Zhao et al., 2019; Li et al., 2021.

Subgroup analyses showed significant increases in UUA levels when FA was used (SMD = 5.22, p = 0.004), in multisystem disease models (SMD = 6.35, p < 0.0001), and in Km mice (SMD = 2.90, p < 0.0001). These subgroup analyses partially account for the heterogeneity observed (Table S1).

Sensitivity analysis

Leave-one-out sensitivity analyses were conducted for all outcomes to observe whether the exclusion of individual studies significantly altered the effect sizes, thus assessing the stability of the results. Throughout these analyses, no significant changes in heterogeneity were observed across all measured outcomes in the meta-analyses (Figs. S1–S6).

Publication bias

To assess publication bias, funnel plots were generated for the overall effects of polyphenolic compounds on SUA and UUA, as well as for the specific effects of RES, CGA, FA, PU, and BER on SUA (Figs. S7–S13). Egger’s tests were conducted using R software, revealing significant publication bias for the overall impact of polyphenolic compounds on SUA and for the effects of RES on SUA, with p < 0.0001 for both.

Discussion

Summary of results

A comprehensive synthesis of data from 49 studies reveals that the five selected polyphenolic compounds significantly reduce SUA levels and enhance UUA excretion in various animal models in comparison to control groups. These findings suggest that these specific polyphenols are effective in lowering uric acid levels, likely through mechanisms associated with increased renal excretion of uric acid. Moreover, subgroup analyses consistently indicate that these polyphenolic compounds decrease SUA levels across different disease models, suggesting that their uric acid-lowering effects are broadly applicable regardless of the underlying disease condition. However, the study exhibits significant heterogeneity. According to the subgroup analyses, variables such as the animal model used, dosage, duration of intervention, mode of administration, and disease model may contribute to this heterogeneity. Notably, subgroup analysis based on animal breeds revealed marked differences in the effects of some polyphenols, which could be attributed to genetic variations affecting uricase activity and uric acid metabolism among different breeds. Currently, experimental mouse models are primarily genetically induced or environmentally induced. Genetically induced models are limited by low survival rates and variable comparability due to complications, while environmentally induced models present challenges in direct comparison due to technical issues, such as variations in doses of inhibitors, induction periods, administration methods, and the measurement of serum uric acid concentrations. Such disparities may lead to inevitable heterogeneity in factors not related to the human condition (Lu et al., 2019). Additionally, individual meta-analyses were conducted to assess the specific effects of each polyphenolic compound (namely RES, CGA, FA, PU, and BER) on reducing SUA levels in animal models. The results confirm that each compound effectively reduces SUA levels.

In preclinical studies, the efficacy of RES in reducing SUA levels has been consistently affirmed. It was previously hypothesized that RES might reduce uric acid production through the inhibition of XO activity (Honda et al., 2014; Zhou et al., 2023). However, recent findings suggest that RES may also promote the enhanced excretion of UUA. Despite these insights, clinical trials examining the effects of RES continue to yield mixed results. For instance, studies conducted by Bo et al. (2016) and meta-analyses by Abdollahi et al. (2023) have observed no significant uric acid-lowering effects of RES in diabetic patients. Conversely, investigations by Zhou et al. (2023) have documented significant reductions in uric acid levels among individuals with dyslipidemia. This meta-analysis reveals that the urate-lowering effect of RES in animal models may be characterized by a potential “inverted U-shaped” nonlinear relationship. It appears that low to moderate doses of RES could reduce uric acid levels by dual inhibition of both XO and URAT1. The implications of high-dose RES are less clear; however, some studies indicate that such doses might lead to hepatotoxicity, increased oxidative stress, and suppression of natural killer (NK) cells, among other sub-toxic effects (Ferreira et al., 2024; Mendes-Braz et al., 2012; Petit et al., 2016). These adverse outcomes may contribute to the diminished uric acid-lowering effect observed at higher doses. Subgroup analysis within this meta-analysis demonstrates that RES shows pronounced efficacy in Swiss mice, particularly when administered orally in doses ranging from 26 mg/kg to 50 mg/kg over periods of 4 to 8 weeks. This suggests that an optimal dosing regimen for RES might involve oral administration at these doses, adjusted for body surface area (Huang Jihan et al., 2004), over a similar duration. However, the effectiveness of RES varies across biological species, and optimal conditions for its application in humans still require further clinical validation. Currently, the statistical power of dose grouping in these studies is inadequate, and uncertainties remain regarding the extrapolation of doses from animal models to humans. Consequently, the inverted U-shaped relationship identified should be considered a preliminary observation, necessitating additional mechanistic studies for verification.

CGA, primarily found in coffee, has been recognized early on for its beneficial effects in metabolic-related diseases (Clifford, 2000). Several clinical studies and animal experiments have reported its role in conditions such as obesity, hyperlipidemia, and diabetes. One animal study on mice with hyperuricemia found that CGA could protect against elevated levels of UA, BUN, Cr, and reduce inflammation through the inhibition of LPS, IL-1β, and TNF-α among other inflammatory factors and related signaling pathways (Zhou et al., 2021). Another animal study supported CGA’s ability to improve renal damage in hyperuricemic mice (Mehmood et al., 2020). Regarding the mechanisms by which CGA reduces uric acid, our results suggest it may be related to increased UUA excretion. Evidence from another animal study supports that CGA increases intestinal uric acid excretion by affecting the levels of intestinal urate transporters (ABCG2, GLUT9), combating hyperuricemia (Mehmood et al., 2019). An in vitro study showed that CGA lactone present in roasted coffee might inhibit XO activity (Honda et al., 2014). The mechanism of CGA’s uric acid reduction appears to be multi-pathway and multi-target. Subgroup results from this meta-analysis show that CGA’s uric acid-reducing effect is lower at 26 mg/kg to 50 mg/kg compared to 0 mg/kg to 25 mg/kg and 51 mg/kg to 100 mg/kg, which may be due to differences in animal models across groups.

In some studies, FA is considered to have a definitive uric acid-lowering effect, possibly including the inhibition of XO activity and increased expression of urate transport proteins (Lou et al., 2023; Zhang et al., 2022; Zhang et al., 2023). This aligns with our meta-analysis results; however, our subgroup analysis found that FA only showed a significant reduction in serum uric acid levels at doses of 101 mg/kg to 300 mg/kg, suggesting there may be a threshold effect for FA’s impact on SUA. The high heterogeneity (>50%) in FA dose subgroups could also mean that FA’s effects are influenced by other factors, necessitating further research for confirmation. Some studies have found that FA could cause renal damage in conditions like diabetic nephropathy, renal cancer, and chronic kidney disease in rats (Bagul et al., 2012; Cai, 2010), therefore, we do not currently recommend the widespread use of FA in uric acid-lowering drug development.

Current research on PU is relatively limited, but some studies have mentioned that PU exhibits a serum uric acid-lowering effect by upregulating some uric acid-excreting transporters and downregulating some that absorb uric acid (Masuoka, 2021). Our results indicate that PU significantly reduces serum uric acid. However, in the 51 mg/kg to 100 mg/kg subgroup, the effect of PU on reducing serum uric acid was unclear, and this group exhibited high heterogeneity (>50%). An in-depth analysis of the two studies in this subgroup suggests that the variance in results could be due to differences in disease models, animal models, intervention durations, individual differences among animals, and publication bias.

Given the limited number of studies on BER, we only included one study, which does not significantly inform the analysis of BER’s impact on SUA. BER is primarily found in a few plants like rockfoil, which may be one reason for the scarcity of research on this compound.

Strengths and limitations

This research represents the inaugural systematic review and meta-analysis dedicated to examining the influence of polyphenolic compounds on uric acid levels within animal models. It encompassed an extensive literature search and meticulous analysis. The implementation of subgroup and sensitivity analyses illuminated potential sources of heterogeneity and evaluated the influence of individual studies on the aggregate outcomes, thus providing invaluable reference points for subsequent inquiries.

Nevertheless, the quality of the studies incorporated was predominantly subpar, characterized by ambiguous risks linked to the randomization processes, allocation concealment, baseline characteristics, and the blinding of researchers. Such deficiencies likely contribute to the observed heterogeneity of research findings. While these limitations could potentially exaggerate effect sizes, the consistency of results across varied models and compounds lends credibility to their biological feasibility. We recommend exercising caution in the interpretation of these findings and advocate for enhanced reporting standards in preclinical research. Both the Egger test and funnel plots revealed indications of publication bias, which may compromise the reliability of the uric acid-lowering effects attributed to polyphenolic compounds and could potentially skew the outcomes of this meta-analysis. This optimistic estimation of effect size necessitates validation through extensive, large-scale preclinical trials in the future. During the data extraction phase, the prevalence of studies presenting data in graphical formats led to unavoidable inaccuracies in manual data extraction. Additionally, the small sample sizes prevalent in the included studies undermine the reliability of both the meta-analysis and subgroup analysis results. Although the duration of interventions accounts for some heterogeneity, other potential variables—such as undisclosed rearing conditions, variations in genetic backgrounds of different strains, and disparities in the isolated compounds and their natural sources—remain to be thoroughly investigated through standardized reporting in future research endeavors. The subgroup analyses conducted in this study were constrained by the reporting dimensions of the original data. We advocate for future animal studies to adhere to the STAIR guidelines for standardized reporting of intervention details to facilitate a more precise attribution of heterogeneity (Savitz & Fisher, 2007). Additionally, the lack of specificity regarding the type of disease in animal models introduced further heterogeneity and led to inquiries into whether the application of the five polyphenols was influenced by disease type to some extent. The analysis did not segregate rodent models into distinct groups, which heightened heterogeneity and posed significant challenges in discussing factors influencing the effects of polyphenols; however, this did not appear to markedly affect the overall or subgroup-specific outcomes.

Research Outlook

Extending the findings from animal experiments to clinical applications presents complex and uncertain challenges. Despite the inclusion of various animal models of hyperuricemia in recent studies, significant discrepancies exist between the pathophysiological mechanisms of hyperuricemia in these models and in human conditions. Firstly, rodents possess functional uricase and exhibit a markedly greater capacity for uric acid metabolism compared to humans, potentially leading to an underestimation of drug sensitivity (Jin et al., 2012). Secondly, replicating the impact of comorbid conditions such as metabolic syndrome and chronic kidney disease, which are commonly associated with hyperuricemia in humans, remains a challenge for these models. To improve the predictive accuracy of translational research, future studies should prioritize the use of uricase knockout (Uox-KO) mice or primate models.

The current dosage range of 5–1,000 mg/kg carries significant risks when directly extrapolated to humans. For instance, a mouse dose of 50 mg/kg of RES, corrected for body surface area, corresponds approximately to 4 mg/kg in humans (Chen et al., 2019). However, interspecies differences in bioavailability could result in substantial variances in actual drug exposure. Additionally, the maximum duration of interventions in the reviewed studies was 24 weeks, whereas management of hyperuricemia typically requires lifelong intervention, and data on chronic toxicity are sparse.

In future research, it may be necessary to employ physiological pharmacokinetic models to predict appropriate dosages for human use and to conduct comprehensive toxicological assessments focused on hepatic, renal, and reproductive toxicity to ensure the reliability of clinical translation.

Conclusion

This meta-analysis indicates that polyphenols, including RES, CGA, FA, PU, and BER, may reduce serum uric acid levels in animal models; however, the clinical relevance of these findings remains to be established. Although RES exhibited an inverted U-shaped nonlinear dose–response relationship, the high degree of heterogeneity and methodological flaws—such as small sample sizes, unclear randomization procedures, and potential publication bias—necessitate a cautious interpretation of the results. To overcome these limitations, future research should prioritize: (1) conducting high-quality preclinical studies using standardized protocols to mitigate variability arising from inconsistent experimental designs; and (2) undertaking interspecific pharmacokinetic studies to compare the bioavailability and metabolite profiles of RES between rodents and primates, thereby facilitating dose extrapolations for early human clinical trials.

Supplemental Information

Supplemental Information 1 PRISMA checklist

Supplemental Information 2 Search strategy

Supplemental Information 3 Supplementary figures and table

Supplemental Information 4 Audience

Additional Information and Declarations

Competing Interests

Author Contributions

Data Availability

The authors declare there are no competing interests.

Jianhong Chen conceived and designed the experiments, performed the experiments, analyzed the data, prepared figures and/or tables, authored or reviewed drafts of the article, and approved the final draft.

Boye Zhang conceived and designed the experiments, performed the experiments, analyzed the data, prepared figures and/or tables, authored or reviewed drafts of the article, and approved the final draft.

Zhongzhi Cao conceived and designed the experiments, performed the experiments, analyzed the data, authored or reviewed drafts of the article, and approved the final draft.

Li Yang conceived and designed the experiments, performed the experiments, authored or reviewed drafts of the article, and approved the final draft.

Ye Yuan conceived and designed the experiments, authored or reviewed drafts of the article, funding acquisition; Resources, and approved the final draft.

The following information was supplied regarding data availability:

This is a systematic review/meta-analysis.

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
