# Peer review of "Urate-lowering effects of polyphenolic compounds in animal models: systematic review and meta-analysis"

_PeerJ, doi:10.7717/peerj.19731_

## Round 0.1 · original submission · Major Revisions

**Language Note:** The review process has identified that the English language must be improved. PeerJ can provide language editing services - please contact us at [email protected] for pricing (be sure to provide your manuscript number and title). Alternatively, you should make your own arrangements to improve the language quality and provide details in your response letter. – PeerJ Staff

·

Basic reporting

The paper systematically reviews and meta-analyzes the urate-lowering effects of five polyphenolic compounds—resveratrol (RES), chlorogenic acid (CGA), ferulic acid (FA), purpurogallin (PU), and bergenin (BER)—in animal models. Based on 49 studies and 1,252 experimental animals, the results indicate these compounds significantly reduce serum uric acid (SUA) levels (SMD = -2.33, 95% CI [-2.73, -1.93]) and increase urinary uric acid (UUA) levels (SMD = 2.53, 95% CI [1.38, 3.69]), with notable contributions from each compound. Subgroup analyses reveal consistent efficacy across various disease models, with RES showing a dose-response relationship. While promising, the findings highlight the need for further clinical validation to confirm these effects in humans. The paper suggests these polyphenols may offer potential as alternative therapies for hyperuricemia and associated diseases. Specific comments:
1. While the background is comprehensive, it would be helpful to expand on why these specific polyphenolic compounds (RES, CGA, FA, PU, BER) were chosen and how they compare to similar compounds in the past.
2. Explain why studies with plant extracts containing the compounds were excluded , given that such data might also provide insights into urate-lowering efficacy.
3. Although the SYRCLE tool was applied, none of the studies were rated as low risk for key criteria such as sequence generation. How does this influence the interpretation of results?
4. The subgroup analysis is well done but lacks explanation for the observed differences between animal species (e.g., Km mice vs. Swiss mice). Consider elaborating on why such variations might occur.
5. The high I² values across multiple analyses indicate significant heterogeneity. Could additional subgroup analyses (e.g., intervention durations) help further reduce heterogeneity?
6. RES exhibited a dose-response relationship, but the data presented do not fully support the clarity of this relationship. Can the authors discuss specific dose ranges and their relative efficacy in more detail?
7. The manuscript hints at clinical applications (lines 41–42), but a more detailed discussion is needed on translational aspects, including safety concerns and potential dosage for human trials.

Experimental design

no comment

Validity of the findings

no comment

Additional comments

no comment

·

Basic reporting

The manuscript is written in professional and comprehensible English. However, minor language improvements could enhance the overall fluency. I recommend careful proofreading by a native English speaker familiar with biomedical writing to refine phrasing and improve flow (e.g., some redundancy could be reduced, and transitions between sections could be smoother).

The introduction provides a solid background and justifies the relevance of the study. Nonetheless, the authors could further strengthen the context by discussing previous meta-analyses or systematic reviews addressing natural compounds for urate-lowering effects, if available.

The manuscript follows a logical and acceptable structure according to PeerJ standards. However, some sections, particularly in the Results, could be condensed for better readability. Subgroup analyses are highly detailed, but summarizing some findings would enhance clarity.

Figures are of high quality, relevant, and appropriately labeled. Raw data are made available, complying with PeerJ policies. Nevertheless, figure legends could be more descriptive to allow standalone interpretation without referring to the main text.

Experimental design

The research question is clearly defined, original, and addresses an important gap regarding the urate-lowering effects of isolated polyphenolic compounds in animal models.

Validity of the findings

The sample size across studies is relatively robust for a preclinical meta-analysis. Sensitivity analyses are properly conducted and provide reassurance about the stability of findings.
However:
Heterogeneity is consistently high (I² > 80%). Although subgroup analyses were performed, additional discussion is needed on other potential sources of heterogeneity (e.g., species differences, strain differences, intervention durations).
Publication bias was detected by Egger’s test. A deeper discussion of how this bias might influence the conclusions would be appropriate.

The conclusions are consistent with the presented results. However, given the generally low methodological quality of the included studies (e.g., unclear randomization, allocation concealment, blinding), the authors should frame their conclusions more cautiously and emphasize the need for better-designed future studies.

·

Basic reporting

The manuscript is generally well written in professional English, with good organization and clarity.

Background and rationale are appropriately explained, and literature is sufficiently cited.

Raw data is presented adequately in figures and supplements.

The use of PRISMA guidelines and PROSPERO registration strengthens reporting quality.

Experimental design

The research question is clearly defined and justified, addressing a gap in urate-lowering research from polyphenols.

The inclusion and exclusion criteria for the animal studies are appropriate and well documented.

Risk of bias was assessed using SYRCLE, which is appropriate for animal studies.

Validity of the findings

The findings appear robust, with consistent meta-analytic approaches applied.

Additional comments

Expand discussion on the limitations related to animal model translation to humans.

Recommend a clearer statement of clinical research implications in future directions.

Kindly remove the mention of 'the two researchers' to maintain formal tone.

---

## Round 0.2 · accepted · Accept

Dear Dr. Chen,

Thank you for submitting the revised version of your manuscript. After a thorough evaluation of your revisions by Reviewer #2 and myself, I am pleased to inform you that all reviewer comments have been satisfactorily addressed. Accordingly, your manuscript is now accepted for publication in PeerJ.

Sincerely,
Stefano Menini

·

Basic reporting

As outlined in my previous comment during the last review

Experimental design

As outlined in my previous comment during the last review

Validity of the findings

As outlined in my previous comment during the last review

Additional comments

All the suggestions have been incorporated in the current amendment